# Clinical assessment of pelvic floor and abdominal muscles 3 months post partum: an inter-rater reliability study

Sabine Vesting [1,2] Monika Fagevik Olsen,[1,3,4] Annelie Gutke,[1] Gun Rembeck,[5,6,7] Maria E H Larsson[1,8]

For numbered affiliations see end of article.

**Correspondence to**
Sabine Vesting;
sabine.vesting@gu.se

## ABSTRACT

**Objectives** Evaluation of the inter-rater reliability of clinical assessment methods for pelvic floor muscles and diastasis recti abdominis post partum.

**Design** A multicentre inter-rater reliability study.

**Setting** Three primary care rehabilitation centres in Sweden.

**Participants** A total of 222 participants were recruited via advertising at Swedish maternity care units and social media. Eligibility for participation included female gender, ≥18 years, at maximum 3 months after childbirth. Exclusion criteria were chronic pelvic girdle pain and/or low back pain and/or pelvic floor tear grade III/IV. At each centre, 2 physiotherapists, with training and experience in pelvic floor assessment, assessed the 222 women according to a standardised protocol in random order.

**Outcome measures** Inter-rater reliability of the assessment of pelvic floor muscle function (involuntary and voluntary contraction and voluntary relaxation) and diastasis recti abdominis (width, depth and bulging).

**Results** Vaginal palpation of maximal voluntary contraction revealed a kappa value of 0.69 (95% CI 0.62 to 0.76). Assessments of involuntary contraction and voluntary relaxation yielded inconsistent results, with slight-to-moderate weighted kappa values ranging from 0.10 to 0.51. After 2 months of training in applying this method, diastasis recti abdominis width measured at the umbilicus by calliper yielded an intraclass correlation coefficient value of 0.83 (95% CI 0.76 to 0.87). Assessments of diastasis recti abdominis depth and bulging showed moderate kappa values, with reservation for some inconsistency between the centres.

**Conclusions** Vaginal palpation of pelvic floor muscle strength is a reliable method for the postpartum muscle assessment. Additional research is needed to identify reliable assessment method for other pelvic floor muscle functions like involuntary contraction and voluntary relaxation. With some training, a calliper is a reliable instrument for measuring the postpartum diastasis recti abdominis width. This study provides novel thoughts about how to measure diastasis recti abdominis depth and bulging.

**Trial registration number** NCT03703804.

## INTRODUCTION

Both the pelvic floor and the abdominal muscles are part of the stabilisation system for

## Strengths and limitations of this study

► This study provides a detailed description of clinically applicable assessment methods for postpartum muscle assessment in primary care.

► This study is raising novel thoughts about the question how diastasis recti abdominis depth and bulging could be assessed post partum.

► The need of training and experience in the clinical assessment of the diastasis recti abdominis was underestimated before start of the study.

► The used assessment methods were not validated, for example, by comparing the results with technical devices like ultrasound and/or pressure measurement.

► The assessment methods were tested at three primary care rehabilitation centres, adding training and experience and maybe even continuity as important aspects for a reliable assessment.

the pelvic and spine[1] as well as the continence system.[2] Pregnancy and childbirth cause alterations in these muscle groups. During a vaginal delivery, the pelvic floor muscles overstretches up to three times.[3] It takes approximately 6 months until muscles, nerves and the connective tissue are recovered from a vaginal delivery.[4 5] Childbirth is associated with pelvic floor traumas, such as perineal tears and levator ani injuries, which can lead to incontinence, pelvic organ prolapse and decreased quality of life.[6 7] Another postpartum muscle alteration is a persistent separation of the two parts of the rectus abdominis, termed a diastasis recti abdominis. At 12 months post partum, 33% of women exhibit a diastasis recti abdominis greater than the width of two fingers.[8] A diastasis recti abdominis is reportedly correlated with impaired quality of life, negative body image and abdominal pain.[9 10]

Women are increasingly seeking help and advice regarding postpartum muscle alterations from physiotherapists at primary healthcare centres. Physiotherapists use various methods to assess the pelvic floor

muscles and diastasis recti abdominis after pregnancy,[11 12] however, there is currently no gold standard. Clinically, vaginal palpation and observation are common methods to assess different pelvic floor muscle functions after childbirth. The International Continence Society defined that the pelvic floor muscle can be involuntarily and voluntarily contracted and can also be voluntarily relaxed.[13] However, there are no standardised assessment protocols or rating scales in the postpartum care, yet. One study reported the use of a Delphi scheme to identify the optimal protocol for assessing different pelvic floor muscle functions and tested their inter-rater reliability,[14] but these assessments were not performed in postpartum women.

Ultrasound assessment is the most reliable and valid method for measurement of the diastasis recti abdominis width.[15] However, most women who are concerned about their diastasis recti abdominis seek help at primary healthcare centres, where ultrasound is seldom available. About 96% of American physiotherapists specialised in women's health assess the diastasis recti abdominis width using the finger-width method,[8] which is imprecise due to finger-width variations[16] and has weaker inter-rater reliability than instrumental assessment methods.[15] Less than 2% of American physiotherapists use callipers for the assessment of postpartum women. Diastasis recti abdominis width assessment using a calliper is reported to be nearly as accurate as ultrasound assessment,[15 17] although the inter-rater reliability of this method has been tested in just one study.[18]

An experimental study suggests that the tendon between the two parts of the rectus abdominis—the linea alba—has less tension in a curl-up movement in women with diastasis recti abdominis.[19] These findings of Lee and Hodges[19] were strengthened by a further study measuring the tension and stiffness in the linea alba.[20] Both research groups argue that the tension in the linea alba is an important factor for the load transfer in the core and might be one of the key components to explain why women with diastasis recti abdominis experience functional impairments. However, these studies were experimental, and their hypothesis is not validated yet. The assessments in these studies were performed by ultrasound[19] and shear-wave elastography.[20] For clinical practice, there are currently no reliable and clinically applicable assessment methods or rating scales for linea alba tension or stiffness. Hypothetical, the tension in the linea alba could be palpated, the inability to control intra-abdominal pressure could be observed as diastasis recti abdominis bulging.[21]

In the present study, we aimed to evaluate the inter-rater reliability of different aspects of the clinical assessment of pelvic floor muscle and diastasis recti abdominis using observation, callipers, and palpation at 3 months post partum.

## MATERIAL AND METHODS
### Study design and subjects
In total, 222 women from the Region Västra Götaland, Sweden were included to this reliability study. Based

on the guidelines of Koo and Li, we aimed to assess at least 30 participants at each centre.[22] Assessments were conducted at three primary care rehabilitation centres. All included women gave their written informed consent to participate. This study is in line with the Declaration of Helsinki and the Standards for Reporting Diagnostic accuracy studies checklist was used to report this study about assessment methods.

The women were invited to participate via advertising at antenatal and childcare centres, and social media. Inclusion criteria were age of ≥18 years, vaginal delivery or caesarean section within the past 3 months, and the ability to understand and respond in Swedish. Exclusion criteria were chronic pelvic girdle pain and/or low back pain (defined as self-reported pelvic or low back pain for over 3 months, not related to the last pregnancy) and/or pelvic floor tear grade III/IV.

The participants were contacted and booked for assessment at one of the three primary care rehabilitation centres in the Västra Götaland region between 8 and 12 weeks after giving birth. Assessments were performed by six physiotherapists—two at each primary care rehabilitation centre. Prior to the assessments, the participants completed a questionnaire about their age, body mass index (BMI), mode of delivery, number of delivered children, self-reported pelvic floor tears, most recent baby's birth weight and the birth weights of previous children (if applicable).

The assessing physiotherapists had each completed a 4-day (or longer) course in pelvic floor muscle assessment and treatment methods. The physiotherapists at primary care rehabilitation centre 1 had 2 and 9 years' experience in pelvic floor muscle assessment at start of the study, the physiotherapists at primary care rehabilitation centre 2 had 1 and 3 years' experience and the physiotherapists at primary care rehabilitation centre 3 had both 1-year experience in pelvic floor muscle assessment at start of the study. All six physiotherapists work at primary care rehabilitation centres, part time or full time with women's health. They had experience in palpating the diastasis recti abdominis with the finger-width method. However, similar to the low numbers of American physiotherapists using the calliper,[11] calliper measurement was new to all raters as it is unusual in Sweden and diastasis recti abdominis assessment is not part of the Swedish physiotherapy education. Therefore, all included physiotherapists were novices at measuring the diastasis recti abdominis by calliper, and on using the rating scales for depth and bulging. During the design phase of this study, 4 hours of training in diastasis recti abdominis measurement was planned.

Due to an insecurity among the physiotherapists in assessing and measuring the diastasis recti abdominis in the beginning of the study, the first 2 months of the study (63 measurements) were used for training and validation of the right technique for using the calliper and assessing depth and bulging. After these 2 months, the physiotherapists at all centres underwent additional training,

comparing their results and discussing their technique. The whole validation process and the final assessment protocol after validation is available in online supplemental file 1.

### Patient and public involvement

Patients and the public were not actively involved in the initial design of the study. However, the application of the calliper was tested in a pilot trial and the opinions of the participants about the application were considered for the design of the final assessment protocol (online supplemental file 1).

### Clinical assessment of the pelvic floor muscles

The pelvic floor muscles were assessed with the participant in supine position on a flat bench, with the legs flexed and slightly abducted. The participants had a pillow under the head. The assessed pelvic floor muscle functions were involuntary and voluntary contraction (by observation and vaginal palpation) and maximal voluntary contraction, pelvic floor muscle endurance and voluntary relaxation (by vaginal palpation). The detailed assessment protocol is shown in online supplemental file 2.

### Clinical assessment of the diastasis recti abdominis

The assessment of the diastasis recti abdominis was conducted in supine position on a flat bench, without a pillow. The participants were in hook-lying position with their arms resting at their sides. The physiotherapists assessed diastasis recti abdominis width using an electronic digital calliper (150 mm, carbon fibre, accuracy ±0.2 mm, 24 se Sverige AB, Kalmar, Sweden). Diastasis recti abdominis depth and bulging were assessed by observation and palpation. All final measures or ratings were results of single assessments to avoid a fatigue effect. The stepwise explanation of calliper application and assessment of diastasis recti abdominis is explained in online supplemental file 1.

On completion of the assessment, the participants rested for 30 minutes in a sitting or lying position. After the 30 min rest, the second physiotherapist conducted the same assessment as described above. The two physiotherapists at each site were assessing in random order, were blinded to each other's findings and not allowed to talk about their assessments. The physiotherapist were also blinded to the participants background information.

### Statistical analysis

Statistical analyses were performed using IBM SPSS statistical package V.25 . Descriptive statistics are presented as mean and SD for ratio data, and as number and percentage for nominal and ordinal data. To calculate statistically significant differences between the three rehabilitation centres, we used the one-way analysis of variance test for ratio data, and the Kruskal-Wallis test for ordinal data and $\chi^2$ test for categorical data. A $p \leq 0.05$ was regarded as statistically significant.

All pelvic floor muscles functional measures were rated on ordinal scales, except for pelvic floor muscle endurance and involuntary contraction by palpation. Diastasis recti abdominis depth was also rated on an ordinal scale. Ratings on ordinal scales were evaluated by Cohen's weighted kappa values. Pelvic floor muscle endurance, involuntary contraction by palpation and diastasis recti abdominis bulging were rated on nominal scales, and these ratings were evaluated by Cohen's kappa values. For interpretation of kappa values, we used the categories of Landis and Koch: <0.2, slight; 0.21–0.40, fair; 0.41–0.60, moderate; 0.61–0.80, substantial and 0.81–1.0, almost perfect agreement.[23] Percentage agreement was calculated and presented for all nominal and ordinal data, and <60% agreement was defined as faulty agreement.[24]

For assessment of diastasis recti abdominis width, a continuous scale (in mm) was used. To evaluate the interrater reliability of the assessments on a continuous scale, we calculated the intraclass correlation coefficient (ICC) and 95% CI. ICC values were calculated in SPSS based on absolute agreement and a two-way mixed effects model. ICC values of <0.50 indicate poor reliability, 0.50–0.75 indicate moderate reliability, 0.75–0.90 good reliability and values of >0.90 indicate excellent reliability.[22]

To further evaluate reliability, we calculated the SE of measurements ($SEM = SD \times \sqrt{(1 - ICC)}$), which represent the typical error in a single measurement and the minimal detectable change ($MDC = SEM \times 1.96 \times \sqrt{2}$). For calculation of the SEM, we used the SD from the scores of all subjects. SEM and MDC values are presented in mm.

## RESULTS

A total of 222 women were assessed, with measurements conducted from September 2018 through February 2020. The mean age of the participants was 33.1 years (SD ±3.3) and the majority of women (61%) had delivered one child (table 1). The participating women at primary care rehabilitation centre 3 were significantly younger than the women at primary care rehabilitation centre 2 and had significantly more children than the participants at primary care rehabilitation centres 1 and 2.

### Clinical assessment of the pelvic floor muscles

The evaluation of maximal voluntary contraction showed substantial agreement (weighted kappa value, 0.69 (95% CI 0.62 to 0.76]), and assessment of pelvic floor muscle endurance showed moderate agreement (kappa value, 0.49 (95% CI 0.37 to 0.61)) (table 2). Seven participants (3.3%) were excluded from the analyses of maximal voluntary contraction and pelvic floor muscle endurance due to incorrect pelvic floor muscle contraction (straining). Assessment of voluntary contraction by observation showed moderate agreement, with a weighted kappa of 0.45 (95% CI 0.28 to 0.62). Among all assessments, about 89% were rated as 'perineal inward movement'.

**Table 1** Characteristics of the participating women at 3 months post partum (n=222)

|  | Total (n=222) Mean±SD | Primary care rehabilitation centre 1 (n=90) Mean±SD | Primary care rehabilitation centre 2 (n=103) Mean±SD | Primary care rehabilitation centre 3 (n=29) Mean±SD | P value* |
|---|---|---|---|---|---|
| Age in years | 33.1±3.3 | 32.6±3.5 | 33.8±2.9 | 32.0±3.8 | **0.01** |
| BMI | 24.5±3.0 | 24.4±3.1 | 24.3±2.9 | 25.1±3.2 | 0.42 |
| Neonatal birth weight | 3574.5±507.0 | 3517.7±533.7 | 3604.7±489.2 | 3641.1±484.0 | 0.59 |
|  | n (%) | n (%) | n (%) | n (%) |  |
| Delivery mode |  |  |  |  | 0.48 |
| C-section | 28 (13) | 9 (10) | 16 (16) | 3 (10) |  |
| Vaginal delivery | 194 (87) | 81 (90) | 87(85) | 26 (90) |  |
| No of children |  |  |  |  | **<0.01** |
| 1 | 137 (61) | 51 (57) | 75 (73) | 11 (30) |  |
| 2 | 74 (33) | 33 (37) | 26 (25) | 15 (50) |  |
| 3 | 9 (4) | 7 (6) | 1 (1) | 1 (3) |  |
| 4 or more | 3 (1) | 0 (0) | 1 (1) | 2 (7) |  |
| Self-reported perineal tear |  |  |  |  | 0.33 |
| No tear | 54 (24) | 24 (27) | 23 (22) | 7 (24) |  |
| First-degree perineal tear | 52 (23 | 16 (18) | 24 (23) | 12 (41) |  |
| Second-degree perineal tear/episiotomy | 74 (33) | 33 (38) | 35 (34) | 6 (20) |  |

Bold values: P<0.05 indicates a significant difference.
*P value by one-way ANOVA test for ratio data, Kruskal-Wallis test for ordinal data and $\chi^2$ test for categorical data.
ANOVA, analysis of variance; BMI, body mass index.

The assessment of involuntary contraction by observation exhibited slight agreement (weighted kappa value, 0.10 (95% CI –0.02 to 0.22)). About 70% of participants were rated as 'downward movement', and 9%–11% as upward movement. Fair-to-moderate kappa values were found for evaluation of involuntary contraction

**Table 2** Results of the clinical assessment of pelvic floor muscles via observation and vaginal palpation

| Parameters | Total group (n=222) Kappa (95% CI) | PA % | Primary care rehabilitation centre 1 (n=90) Kappa (95% CI) | PA % | Primary care rehabilitation centre 2 (n=103) Kappa (95% CI) | PA % | Primary care rehabilitation centre 3 (n=29) Kappa (95% CI) | PA % |
|---|---|---|---|---|---|---|---|---|
| Voluntary contraction |  |  |  |  |  |  |  |  |
| Observation | 0.45 (0.28 to 0.62) | 90 | 0.40 (0.11 to 0.70) | 91 | 0.55 (0.33 to 0.77) | 90 | 0.29 (–0.13 to 0.70) | 86 |
| Vaginal palpation |  |  |  |  |  |  |  |  |
| MVC* | 0.69 (0.62 to 0.76) | 71 | 0.70 (0.58 to 0.82) | 77 | 0.59 (0.46 to 0.71) | 67 | 0.67 (0.50 to 0.84) | 62 |
| PFM Endurance* | 0.49 (0.37 to 0.61) | 74 | 0.68 (0.53 to 0.83) | 84 | 0.27 (0.09 to 0.45) | 65 | 0.55 (0.20 to 0.90) | 83 |
| Involuntary contraction |  |  |  |  |  |  |  |  |
| Observation | 0.10 (–0.02 to 0.22) | 57 | 0.20 (–0.01 to 0.42) | 77 | 0.11 (–0.05 to 0.27) | 48 | 0.38 (0.10 to 0.67) | 68 |
| Vaginal palpation | 0.51 (0.37 to 0.65) | 85 | 0.26 (–0.04 to 0.56) | 87 | 0.47 (0.25 to 0.69) | 85 | 0.47 (0.14 to 0.80) | 75 |
| Voluntary relaxation |  |  |  |  |  |  |  |  |
| Vaginal palpation | 0.26 (0.15 to 0.37) | 57 | 0.30 (0.16 to 0.50) | 63 | –0.08 (–0.23 to 0.07) | 45 | 0.56 (0.07 to 1.02) | 89 |

*Reduced number of participants due to incorrect pelvic floor muscle contraction (straining): total group (n=215), primary care rehabilitation centre 1 (n=88), primary care rehabilitation centre 2 (n=98), primary care rehabilitation centre 3 (n=29).
MVC, maximal voluntary contraction; PA, percentage agreement; PFM, pelvic floor muscle.

**Table 3** Diastasis recti abdominis width at 3 months post partum (in mm) measured with a calliper

| | Above the umbilicus Mean±SD 95% CI | At the umbilicus Mean±SD 95% CI | Below the umbilicus Mean±SD 95% CI |
|---|---|---|---|
| Total group (n=159) | 22.0±7.4 (20.9 to 23.2) | 25.9±7.2 (24.8 to 27.1) | 19.6±7.3 (18.4 to 20.7) |
| Primipara (n=97) | 20.73±6.0 (19.5 to21.9) | 24.3±5.7 (23.1 to 25.4) | 19.0±9.9 (17.6 to 20.4) |
| Multipara (n=62) | 24.1±8.9 (21.9 to 26.4) | 28.7±8.5 (26.5 to 30.8) | 20.6±7.9 (18.6 to 22.6) |
| Primary care rehabilitation centre 1 (n=61) | 19.8±5.3 (18.5 to 21.2) | 24.1±6.5 (22.5 to 25.8) | 14.8±5.8 (13.3 to 16.2) |
| Primipara (n=35) | 18.7±4.5 (17.2 to 20.3) | 22.6±5.1 (20.9 to 24.4) | 13.7±5.1 (12.0 to 15.5) |
| Multipara (n=26) | 21.3±6.0 (18.8 to 23.9) | 26.1±7.6 (23.1 to 29.5) | 16.2±6.4 (13.5 to 18.8) |
| Primary care rehabilitation centre 2 (n=69) | 20.6±4.6 (19.5 to 21.7) | 24.7±5.2 (23.5 to 25.9) | 21.1±4.6 (20.0 to 22.2) |
| Primipara (n=51) | 20.4±4.5 (19.1 to 21.7) | 24.3±4.6 (23.0 to 25.6) | 21.3±4.2 (20.1 to 22.5) |
| Multipara (n=18) | 21.3±4.9 (18.9 to 23.8) | 25.9±6.6 (22.6 to 29.1) | 20.4±5.8 (17.6 to 23.3) |
| Primary care rehabilitation centre 3 (n=29) | 29.9±10.9 (25.8 to 34.1) | 32.8±9.0 (29.3 to 36.2) | 26.1±9.0 (22.7 to 29.6) |
| Primipara (n=11) | 28.6±9.4 (22.3 to 34.9) | 29.5±9.2 (23.3 to 35.7) | 25.0±11.0 (17.6 to 32.4) |
| Multipara (n=18) | 30.7±11.9 (24.8 to 36.6) | 34.8±8.5 (30.5 to 39.0) | 26.9±7.8 (23.0 to 30.7) |

by palpation. Over 80% of participants were rated as 'absence of correct contraction'.

Ratings of voluntary relaxation showed large variations between different primary care rehabilitation centres, with weighted kappa values ranging from −0.08 to 0.56. The application of the scale significantly differed between primary care rehabilitation centre 3 and primary care rehabilitation centres 1+2. At primary care rehabilitation centre 3, the physiotherapists rated 25 of 29 assessments as showing complete voluntary relaxation. In contrast, at primary care rehabilitation centres 1+2, 10–12% of participants were rated as absent voluntary relaxation, 66% as partly relaxed and 20%–24% as complete voluntary relaxation. Primary care rehabilitation centre 2 showed a negative kappa value of −0.08, indicating an agreement worse than expected or no agreement.[24]

### Clinical assessment of the diastasis recti abdominis
Diastasis recti abdominis width, depth and bulging were assessed in 159 women. The mean diastasis recti abdominis widths at 3 months post partum was 25.9 mm at the umbilicus (table 3). The diastasis recti abdominis widths measured at rehabilitation centre 3 were significantly wider at all three measure points compared with rehabilitation centres 1 and 2 (p<0.01). The diastasis recti abdominis widths measured at the umbilicus was significantly wider in women with more than one child at

rehabilitation centre 1 (p=0.04) but not at rehabilitation centre 2+3.

The measurement of diastasis recti abdominis width with a calliper showed good reliability when measured at the umbilicus and 4.5 cm below the umbilicus, and moderate reliability at 4.5 cm above the umbilicus (table 4). For the total group, the SEM was between 4.05 and 4.75 mm, and the MDC was 11.23–13.17 mm. Subanalysis of the different primary care rehabilitation centres revealed two negative outliers. At primary care rehabilitation centre 2, assessment at 4.5 cm below the umbilicus showed an ICC value of 0.51 (95% CI 0.20 to 0.70), which is at the lower boundary of the definition for moderate reliability. Assessment at 4.5 cm above the umbilicus at primary care rehabilitation centre 3 showed much lower ICC values compared with the other values. An ICC value of 0.40 indicates low reliability.

The assessment of diastasis recti abdominis depth showed fair-to-moderate weighted kappa values ranging from 0.34 to 0.43 (table 5). In the assessment of the diastasis recti abdominis depth, 21% were assessed as 'good resistance at all points', 67% as 'resistance in the depth' and 12% as 'bottomless'. Assessment of diastasis recti abdominis bulging in the three-step sit-up test showed moderate agreement (kappa value, 0.51 (95% CI 0.29 to 0.73)). The diastasis recti abdominis bulging was rated in

**Table 4** Results of the clinical assessment of diastasis recti abdominis width with a calliper (in mm)

| Parameters | Total group (n=159) ICC (95% CI) SEM; MDC | Primary care rehabilitation centre 1 (n=61) ICC (95% CI) SEM; MDC | Primary care rehabilitation centre 2 (n=69) ICC (95% CI) SEM; MDC | Primary care rehabilitation centre 3 (n=29) ICC (95% CI) SEM; MDC |
|---|---|---|---|---|
| Above the umbilicus | 0.73 (0.63 to 0.80) 4.75; 13.17 | 0.78 (0.63 to 0.87) 5.32; 14.75 | 0.60 (0.36 to 0.75) 3.46; 9.59 | 0.40 (−0.32 to 0.72) 8.30; 23.01 |
| At the umbilicus | 0.83 (0.76 to 0.87) 4.05; 11.23 | 0.85 (0.75 to 0.91) 3.29; 9.12 | 0.62 (0.39 to 0.77) 4.34; 12.03 | 0.82 (0.61 to 0.91) 4.93; 13.66 |
| Below the umbilicus | 0.80 (0.72 to 0.85) 4.40; 12.20 | 0.75 (0.58 to 0.85) 3.64; 10.09 | 0.51 (0.20 to 0.70) 4.03; 11.17 | 0.74 (0.43 to 0.88) 6.16; 17.07 |

ICC, intraclass correlation coefficient; MDC, minimal detectable change; SEM, SE of measurements.

all participants, even those who were not able to perform a complete sit-up. According to Lo and Candido, the diastasis recti abdominis bulging occurs on exertion.[25] Our hypothesis was that all participants did their individual maximal exertion to perform a sit-up even if it was not resulting into a complete sit-up. However, the assessing physiotherapist had the rating option 'cannot assess' in case of insecurity. The values at primary care rehabilitation centre 2 and 3 were lower than at primary care rehabilitation centre 1 and showed wide ranges in the 95% CI. Among the participants, about 81% were rated as 'no bulging', 12% as 'diastasis recti abdominis bulging' and 7% as 'cannot assess'.

## DISCUSSION

The main findings of this study are that physiotherapists managing women's health in primary care have reliable methods available to assess voluntary pelvic floor muscle contraction by vaginal palpation and measure diastasis recti abdominis width by calliper 3 months post partum. On the other hand, the assessment of involuntary pelvic floor muscle contraction by observation and voluntary pelvic floor muscle relaxation had kappa values with slight-to-fair agreement. Data with such low agreement are not useful for clinical practice or research.[24] Further investigations are needed to improve the clinical applicability and reliable assessment of these functions.

### Clinical assessment of the pelvic floor muscles

Our present results showed weighted kappa values of 0.59–0.70 for the assessment of maximal voluntary contraction, which are higher compared with previous studies.[26–28] One explanation could be differences between the rating scales used for this muscle function. For example, in two prior studies,[26 27] only the squeezing and not the lifting component of the contraction was rated. Devreese et al discussed the importance of the lifting component of pelvic floor muscle strength, showing that incontinent women showed less inward movement than continent women.[29] Furthermore, the differences to our results could also be related to differences in the study population and smaller sample sizes in previous studies.

The assessment of pelvic floor muscle endurance showed moderate reliability. However, there were several inconsistencies in the pelvic floor muscle endurance data. Assessment of pelvic floor muscle endurance at primary care rehabilitation centre 2 showed only fair reliability, with a kappa value of 0.27 (95% CI 0.09 to 0.45). Our present results are lower compared with the findings of Devreese et al.[29] Notably, in the study of Devreese et al, a contraction longer than 10 s was rated as positive rather than a contraction longer than 30 s, as in the present study. Additionally, their study population was older and not 3 months post partum. It may be more difficult to assess the exact point of time when the contraction is

**Table 5** Results of the clinical assessment of diastasis recti abdominis depth and bulging via observation and palpation

| | Total group (n=159) Kappa (95% CI) | PA % | Primary care rehabilitation centre 1 (n=61) Kappa (95% CI) | PA % | Primary care rehabilitation centre 2 (n=69) Kappa (95% CI) | PA % | Primary care rehabilitation centre 3 (n=29) Kappa (95% CI) | PA % |
|---|---|---|---|---|---|---|---|---|
| Depth | 0.43 (0.29 to 0.56) | 69 | 0.37 (0.15 to 0.59) | 70 | 0.34 (0.11 to 0.58) | 71 | 0.36 (0.04 to 0.69) | 62 |
| Bulging* | 0.51 (0.29 to 0.73) | 88 | 0.77 (0.52 to 1.02) | 94 | 0.35 (0.04 to 0.66) | 83 | 0.36 (−0.16 to 0.88) | 88 |

*The physiotherapists had the rating option of 'cannot assess', and these assessments were excluded: total group (n=137); primary care rehabilitation centre (n=52); primary care rehabilitation centre 2 (n=59) and primary care rehabilitation centre 3 (n=26).
PA, percentage agreement.

subsiding in a longer time period. It is also possible that postpartum women have, on average, a weaker contraction, making it more difficult to assess pelvic floor muscle endurance. Further research is needed to decide whether a pelvic floor muscle endurance of 10 s or 30 s is more clinically relevant.

Voluntary contraction via observation showed fair-to-moderate weighted kappa values. A prior MRI study reported that the average inward movement of the perineum is about 1 cm while sitting, and it is more than 2 cm in the supine position, according to Kegel in 1952 as cited in Bo et al.[30] It could be assumed that this large movement would be easy to observe, and a higher kappa value was expected. A previous study reported high inter-rater reliability in the observation of inward perineum movement.[14] Correspondingly, another study showed that inward perineal movement could be observed with a kappa value of 0.91 among continent women, and 0.93 among incontinent women.[29]

As factors other than pelvic floor muscle strength may contribute to urinary leakage post partum,[31] it is important to assess other aspects of pelvic floor muscle function. Unfortunately, in our present study, we did not find a reliable method for assessing involuntary contraction by observation, and we demonstrated inconsistent findings for the assessment of involuntary contraction by palpation. Accordingly, another study reported only fair inter-rater reliability for the assessment of involuntary contraction by observation and palpation.[14] Further studies are needed to develop improved methods for the assessment and rating of these pelvic floor muscle functions in clinical practice.

When prescribing postpartum pelvic floor muscle training, it must be considered that some women have overactive, and possibly painful pelvic floor muscles.[32] It remains unclear whether women with hypertonic pelvic floor muscles should be advised to do pelvic floor muscle training. In clinical practice, physiotherapists recommend an individualised approach. Our present results showed that the rating of voluntary relaxation had slight-to-fair inter-rater reliability. Slieker-ten Hove et al[14] reported similar findings. Another study used a five-step rating scale for relaxation after contraction, and reported a correlation of 0.34 between two raters.[32] Even, Slieker-ten Hove et al recommend the addition of more rating steps to the scale, for example, incomplete relaxation in their discussion. Regardless of whether women with hypertonic pelvic floor muscles need more support in pelvic floor muscle training or the recommendation of no pelvic floor muscle training, there remains a need for better methods of assessing this condition.

### Clinical assessment of the diastasis recti abdominis

Our results from the present study showed moderate-to-good reliability in measuring diastasis recti abdominis width using a calliper, after 2 months of training and calibrating of the method. The characteristics of the diastasis recti abdominis at 3 months post partum measured with the calliper at rehabilitation centre 1+2 were comparable with the diastasis recti abdominis characteristics after pregnancy measured by ultrasound,[33] indicating these values are true values for this population. However, the 2 months of training and calibrating of the method via discussion and additional training were important components of this study. One bias identified in our additional training was that the accurate head lift of just 2–3 cm was an important factor. This observation is supported by the study of Mota et al,[34] which showed that the distance between the two parts of the rectus abdominis decreased during a sit-up movement.

There was some inconsistencies in the data, with two outliers at primary care rehabilitation centres 2+3. The SEM and MDC were higher at primary care rehabilitation centre 3 compared with primary care rehabilitation centres 1+2 (table 4). The MDC measured at 4.5 cm above the umbilicus was over 2 cm, raising doubt about whether these results have any clinical relevance. Taking into concern, that the measured diastasis recti abdominis width at primary care rehabilitation centres 3 are significantly wider than at primary care rehabilitation centres 1+2, seen in both primiparous and multiparous women, questions about if it is more difficult to measure a wider diastasis recti abdominis or if the measures are not accurate have to been raised. Comparing the results from rehabilitation centre 3 with the normal values for primiparous women of Mota el al,[33] there is an indication for residual problems with test performance even after retraining. At primary care rehabilitation centre 3, fewer than 30 participants were recruited during the study period of over 1 year. These results indicate that in addition to training and experience, some continuity in measuring the diastasis recti abdominis width with a calliper is necessary for reliable assessment.

The SEM in our present study was about 4–5 mm and the MDC was about 11–13 mm, which is higher than in the comparable study of Benjamin et al.[18] However, just two participants showed an inter-recti distance greater than 22 mm in their study, in our study the mean diastasis recti abdominis width was 25.9 mm. Furthermore, Benjamin et al are not discussing how much experience and training the two investigating physiotherapists had in measuring diastasis recti abdominis width with a calliper.

We found fair-to-moderate weighted kappa values for the assessment of diastasis recti abdominis depth. To the best of our knowledge, there is no other study with which to compare these results. About 63%–66% of the assessments were rated as 'resistance in the depth'. The investigating physiotherapists hypothesised that 'resistance in the depth' was felt and assessed as soon as the participants did not activate their deeper abdominal muscles during the head lift. Accordingly, Lee and Hodges[19] described the increased tension in the linea alba caused by activation of the deep abdominal muscles. If this is true, the assessment of diastasis recti abdominis depth is an assessment of the tension in the linea alba due to musculus transversus abdominis activation during a head lift.

Another hypothesis could be that diastasis recti abdominis depth is caused by a reduced stability in the linea alba and a greater laxity in the tissue.[35] A next step could be to correlate the depth with the width—with the hypothesis that transversus abdominis activation increases the width[34] and decreases the depth. However, the results of our study showed just fair-to-moderate inter-rater reliability which makes it questionable if palpation is the right method for the clinical assessment of diastasis recti abdominis depth. Future studies must more precisely define the preactivation of the deep abdominal muscles in the assessment of diastasis recti abdominis depth. Similar considerations have to be raised about abdominal muscle preactivation and diastasis recti abdominis bulging during a sit-up curl. Before implementation of this methods to clinical practice, further validation of the assessments of diastasis recti abdominis depth and bulging, for example, a correlation between ultrasound/shear-wave elastography and palpation, is needed.

## Strengths and limitations

One strength of this study was the large sample size and the quantity of different aspects of muscle function. A comparable study assessing different aspects of pelvic floor muscle function in women with and without pelvic floor disorders included only 41 participants.[14] A review about diastasis recti abdominis assessment methods included studies with 20–106 participants, and these studies only examined diastasis recti abdominis width.[15] Another strength of this study was that we were able to perform the same tests at three different centres in different parts of west Sweden. This makes our results transferable for different physiotherapists using the same methods.

The present study also had several limitations. One was that we lacked access to the participants' delivery records. Thus, the descriptive statistics and the calculate statistically significant differences between the three rehabilitation centres regarding age, BMI, mode of delivery, pelvic floor tearing and highest birth weight were based on self-reported data from the participants. Another issue was the uneven cell distribution seen in over 50% of the rating scales tested in this study. In the literature, it is controversial whether a low kappa value can be explained by uneven cell distribution or low prevalence of a condition.[36]

Another limitation but also a strength of this study was the initial assumption that 4 hours of training in measuring the diastasis recti abdominis width with a calliper would be enough training for the six assessing physiotherapists. Due to this misjudgement of complexity using this method, a total of 63 measurements had to be excluded. On the other hand, many different obstacles, and sources of error for the measurements could be found through this initial phase and the added training. Due to the troubleshooting around the right application and procedure while measuring the diastasis recti abdominis with a calliper, the procedure of this study might not 100% comparable with other studies. There are just a few existing studies measuring by calliper,[17 37–40] several

measurement positions like at rest, in a crunch position and in a modified curl-up were used, no consent about the right application method is found yet. In the training sessions to this study, we found it almost impossible to measure in resting, even if other studies[39 40] described the measurement in resting as reliable methods. At the same time, we registered that the higher the head lift/shoulder lift the less width was measurable compared with at rest which is in line with a study showing that an abdominal crunch is narrowing the inter-recti distance.[34] Our intention was to find an active position without missing the actual width of the diastasis recti abdominis. It would have been even better to validate these calliper measurements against ultrasound measurements. However, the aim of this study was to evaluate clinical applicable methods, which are easy to use and inexpensive tools for daily practice. Validation via ultrasound could be a next step.

## Future research

Regarding the pelvic floor muscles, there remains a need for further research on how to assess and rate involuntary contraction, and voluntary relaxation, in the clinical assessment of women after pregnancy due to the high clinical relevance of this functions.

For the diastasis recti abdominis assessment, the depth and bulging have to be further defined and more knowledge about the behaviour of the linea alba during muscle contraction and exertion is needed. For both pelvic floor muscle and diastasis recti abdominis assessment, we need to investigate the extent to which these values are clinically relevant for postpartum women—for example, if the assessment outcomes are associated with pain and dysfunction. Furthermore, we must determine a cut-off point for diastasis recti abdominis severity relative to pain and dysfunction. It will also be important to determine what training advice should be given to postpartum women based on the results of these examinations.

## CONCLUSION

Women are increasingly requesting assessment of their pelvic floor muscles and diastasis recti abdominis after pregnancy. Our present results revealed moderate-to-substantial reliability for the assessment of maximal voluntary contraction and pelvic floor muscle endurance by vaginal palpation, considering both the lifting and squeezing component of the contraction. Furthermore, diastasis recti abdominis width can be measured by calliper, with an SEM of 4–5 mm and an MDC of about 1.2 cm. However, assessment using this instrument requires some training and experience. More research about the assessment of the involuntary contraction and voluntary relaxation of the pelvic floor muscles, as well as the diastasis recti abdominis depth and bulging is needed, before clinical implementation.

## Author affiliations

[1]Department of Health and Rehabilitation, Unit of Physiotherapy, Institute of Neuroscience and Physiology, Sahlgrenska Academy, University of Gothenburg, Gothenburg, Sweden
[2]Närhälsan Gibraltar Rehabilitation, Gothenburg, Sweden
[3]Department of Physical Therapy and Occupational Therapy, Sahlgrenska University Hospital, Gothenburg, Sweden
[4]Department of Gastrosurgical Research & Education, Sahlgrenska Academy, Gothenburg University, Gothenburg, Sweden
[5]Region Västra Götaland, Research and Development Centre Södra Älvsborg, Borås, Sweden
[6]Region Västra Götaland, Regionhälsan Borås Youth Centre, Borås, Sweden
[7]Primary Health Care, School of Public Health and Community Medicine, Institute of Medicine, Sahlgrenska Academy, University of Gothenburg, Gothenburg, Sweden
[8]Region Västra Götaland Education, Research and Development Primary Health Care, Gothenburg, Sweden

**Acknowledgements** We thank Lisa Altvall, Ulrika Hansson, Anna-Lena Magnberg, Vanesa Rufete Bernal, Ute Jesberg, and Johanna Ekberg, who assessed the 222 women in the current study.

**Contributors** Each author of this paper meets the criteria for authorship. SV, MFO, AG, GR and MEHL designed the study. SV collected the data, analysed and performed a first interpretation of the results, and drafted the article. MFO, AG, GR and MEHL participated in the final interpretation of the results, and critically revised the article for important intellectual content. The final manuscript was seen and approved by all authors.

**Funding** This study was funded by the Local Research and Development Council Gothenburg and Södra Bohuslän (grant number VGFOUGSB-762581) and The Healthcare Board, Region Västra Götaland (grant number VGFOUREG-751101).

**Competing interests** None declared.

**Patient consent for publication** Not required.

**Ethics approval** The study protocol was approved by the Swedish ethical review authority in Gothenburg in April 2018 (Dnr 088–18) and registered with the local data protection service, Närhälsan Gothenburg, in January 2019.

**Provenance and peer review** Not commissioned; externally peer reviewed.

**Data availability statement** Data are available on reasonable request. The dataset from this study is available via a reasonable request to the corresponding author.

**ORCID iD**
Sabine Vesting http://orcid.org/0000-0002-6523-1833

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
