## [Reviewer comments · BMJ Open]

ARTICLE DETAILS

TITLE (PROVISIONAL)	Clinical assessment of pelvic floor and abdominal muscles three months postpartum: An inter-rater reliability study
AUTHORS	Vesting, Sabine; Olsen, Monika Fagevik; Gutke, Annelie; Rembeck, Gun; Larsson, Maria

VERSION 1 – REVIEW

REVIEWER	Nygaard, Ingrid University of Utah School of Medicine, OB/GYN
REVIEW RETURNED	12-Mar-2021

GENERAL COMMENTS	The aim of this study was to assess the inter-rater reliability between 2 physiotherapists at each of 3 sites for clinically assessed pelvic floor muscle and diastasis rectus measures. This is important, as dependence on measures such as ultrasound or MRI limits community-based research and thus generalizability of findings. The investigators assess different aspects of each muscle group and conclude that those measures in widest use, maximal pelvic floor muscle contraction and diastasis recti width, have the highest reliability. In contrast, some other suggested measures of function such as involuntary PFM contraction, have very poor reliability which raises the question of whether this can indeed be assessed clinically. One of the factors the authors cite as a limitation (page 19, line 51) is in my opinion also a major strength of the study: that is, they realized that much more training was needed to enhance reliability of DR measures and thus discarded the initial 63 measures, provided re-training, and then began again. As a result, they provide very useful information in the supplemental materials that will aid future researchers in obtaining these measures. Of some concern is the wide range of reliability for PFM endurance with kappa values ranging from 0.27 to 0.68 amongst sites. This raises the question of whether additional training, similar to that done for the abdominal wall measures, would be helpful in strengthening reliability of this variable. Specific questions follow: 1) Title: Suggest adding “inter-rater” before reliability in title and aims.2) Abstract, results, line 48: Add “at the umbilicus” as other ICC values were not quite as high.3) Keywords: Recommend adding pelvic floor muscles
---

	4) The participating physical therapists had completed at least a 4-day course in pelvic floor muscle assessment and treatment and all had 1-9 years' experience in assessing these muscles. This is important information to place in the abstract, as it speaks to the generalizability of the findings. Consider adding to last sentence of 'participants': "At each center, 2 physiotherapists with training and experience in pelvic floor assessment assessed...." (To decrease words, you could delete "with the ability to understand and respond in Swedish", as this is presented later.) 5) Page 5, line 37: This should be "International Continence Society" 6) Page 7, line 30: Please state the minimum number of days after delivery exams were undertaken (that is, "within 3 months" provides only the maximum). 7) Page 7, line 36: "Plinth" is not a familiar word to me (and the definition in Webster's is not likely what you mean!)—please find a substitute. 8) Methods: Was the order of the PT exams at each site randomly assigned? If not, please state in results whether an order effect was seen, particularly for the modified Oxford scale. 9) Page 11 Table 1, line 30: 3% for center 3 for number of children=1 should be 30% 10) It is true that reliability for diastasis recti width measurement was good (discussion, page 15, lines 30-34). However, Table 3 (page 13) raises the question of whether the measure is accurate. The mean DR width at center 3 was, as you note, significantly wider than other sites, with about a 1-cm difference between center 3 and center 1 for all three locations. This may be due to the difference in population at center 3 (particularly notable is increased proportion of multiparas at center 3) or may be due to residual problems with test performance even after retraining. It would be helpful to perform an exploratory analysis of mean width at each site in just the subgroup with 1 child and another in just the subgroup with 2 children to get a sense of whether the results seem more in line. 11) Page 19, line 37-41: I did not see an analysis of differences according to age, BMI, etc in the results section. 12) Page 20, line 43, conclusion: Probably "requesting" would be a better translation than "demanding". 13) Appendix: This information is very valuable and will be helpful to future researchers and clinicians. The photographs are helpful in understanding how measures were taken. For diastasis recti abdominis bulging, please describe the 3-step sit-up test, rather than simply providing a reference.
--	--

REVIEWER	Theodorsen, N-M University of Bergen
REVIEW RETURNED	29-Mar-2021

GENERAL COMMENTS	Dear authors and editor. This is an interesting study, and I do believe that clinicians worldwide would benefit from reliability studies regarding both pelvic floor function and presence and size of diastasis rectus abdominis. However, as this field of research is based on scant and poor-quality research, it is important that research conducted
--

and published is of high quality and can bring increased and evidence-based knowledge to how we assess both the pelvic floor and diastasis rectus abdominis. It is a joint responsibility we all have; to produce, review and publish articles which improve the healthcare postpartum women receives.

My comments to the manuscript are as follows:

1. Is the research question or study objective clearly defined?

I recommend that it is clearly stated both in the abstract and the article what type of reliability was investigated. Interrater/intrarater reliability or agreement. This is important as you are investigating reliability of assessment by palpation. With regards to diastasis rectus abdominis measurement by palpation is not recommended between raters (1, 2)

2. Are the methods described sufficiently to allow the study to be repeated?

Methods regarding pelvic floor function are sufficiently described. The methods described to measure the diastasis rectus abdominis width is not sufficiently described. It is rather confusing to the reader whether the width has been measured during complete rest or during a headlift. In the existing literature the width of the diastasis rectus abdominis is either measured at rest, with the head resting on a plinth or a pillow, or at activation of the abdominal muscles, including a headlift. As far as I can interpret the assessment protocol of the diastasis rectus abdominis, the width was measured with a relaxed headlift. How can you then be sure that the abdominal musculature is actually at rest? And why do you choose a different measurement protocol to other studies. Please explain the rationale for this further, as it may imply that the results of this study cannot be compared to results from other similar studies (2-6).

On page 8, last paragraph, it is stated that the assessment of the diastasis recti abdominis was conducted with the participants in the same position as described for the assessment of the pelvic floor muscles. It is described that the head rests on a pillow. In the supplement however, it is stated that there is no pillow during the assessment. Please clarify.

It is not described where the participants keep their arms during the assessment of the diastasis rectus abdominis. This is important as different arm positions may contribute to different recruitment of the abdominal muscles.

Regarding the assessment of the pelvic floor, it is described in the supplement that during observation of the pelvic floor muscle contraction, the physiotherapist stood beside the plinth, holding the participants legs...I assume this ought to be supporting the participants legs? This must be clarified.

The rater population ought to be more specified with regards to experience, training, work in this area etc, and explain the rationale for including only novice raters for the diastasis rectus abdominis measurements. Equally for the pelvic floor assessment, the different raters experiences may influence the results (1).

Was the final measure/rating a result of single or repeated measures?

Were the raters blinded to the participants background information (Hawthorne effect)?

3. Are research ethics addressed appropriately?
Is this study in line with the Helsinki declaration? Also include what reporting guidelines were used when writing the manuscript.

4. Are the references up to date and appropriate?
Page 5, 2nd paragraph line 7: Spelling mistake. It should be The International Continence Society (ICS). Not The international incontinence society.
Page 3-4, 1st paragraph: The function of the human core and how the different muscles interact and how this may affect function is under debate. I would recommend modifying and adding other references to the introduction. I also question the use of the reference of Mota et al (2015) when referring to postpartum muscular alterations can lead to feelings of insecurity. They found that women with diastasis rectus abdominis were not more likely to report lumbo pelvic pain than women without diastasis rectus abdominis (7). The rationale to measure diastasis rectus abdominis depth is based on the study by Lee & Hodges (2016). They argue that the tension or depth of the Linea Alba may be used to assess linea alba behaviour in patients with diastasis rectus abdominis. However, they have based these theories on a measurement method which is not tested for validity nor reliability(8). Will the depth of the diastasis rectus abdominis give information on the actual depth, or the behaviour and tension in the connective tissue linea alba? One ought to be careful to base future studies on findings of studies with poor research quality.

5. Are the discussion and conclusions justified by the results? Are the study limitations discussed adequately?
As the authors of this manuscript states there are no validated and clinically applicable assessment methods or rating scales for the parameters of diastasis recti abdominis depth and bulging. I therefore question the validity of the depth measurement in this manuscript. How do the authors know that the raters are measuring the depth of the diastasis, and not the tension in the connective tissue? Physiotherapists in clinical practice do already assess the depth/tension, but we do not know how to interpret the results. I would welcome more information on what you assume you are measuring. When the results of the study show fair to moderate weighted values and the validity of the depth assessment is questionable, I would recommend avoiding conclusions that this is a method which may be used in clinical practice. If clinicians read more into this assessment

method than what the evidence allow, it will certainly not improve the patient care. I do not think this study adds two novel methods and scales for the assessment of the diastasis recti abdominis. More studies are needed to establish why we should measure depth/tension and bulging, and we need more reliable methods than these assessed here.

Conclusion: With some adjustments to the rationale of the study, a more detailed description of the methods applied, and the assessment protocols used, this may prove to be a relevant study for clinicians to increase knowledge on how to measure pelvic floor function and diastasis rectus abdominis. It may also prove some novel knowledge on what and why to the different measurements applied. The results regarding the reliability of the voluntary pelvic floor contraction and the width of the diastasis measurements using callipers confirm already established knowledge. As far as the other measurements the results are too weak to draw any definite conclusion other than the fact that further studies are needed. Despite this, I do think that the findings of the reliability of depth/tension assessments are very interesting!

1. Kottner J, Audigé L Fau - Brorson S, Brorson S Fau - Donner A, Donner A Fau - Gajewski BJ, Gajewski Bj Fau - Hróbjartsson A, Hróbjartsson A Fau - Roberts C, et al. Guidelines for Reporting Reliability and Agreement Studies (GRRAS) were proposed. (1878-5921 (Electronic)).
2. van de Water ATM, Benjamin DR. Measurement methods to assess diastasis of the rectus abdominis muscle (DRAM): A systematic review of their measurement properties and meta-analytic reliability generalisation. *Manual Therapy*. 2016;21:41-53.
3. Boxer S, Jones S. Intra-rater reliability of rectus abdominis diastasis measurement using dial calipers. *Australian Journal of Physiotherapy*. 1997;43(2):109-14.
4. Mota P, Pascoal AG, Sancho F, Bø K. Test-retest and intrarater reliability of 2-dimensional ultrasound measurements of distance between rectus abdominis in women. *The Journal of orthopaedic and sports physical therapy*. 2012;42(11):940.
5. Sperstad JB, Tennfjord MK, Hilde G, Ellström-Engel M, Bø K. Diastasis recti abdominis during pregnancy and 12 months after childbirth: prevalence, risk factors and report of lumbopelvic pain. *British Journal of Sports Medicine*. 2016;50(17):1092.
6. Chiarello CM, McAuley JA. Concurrent validity of calipers and ultrasound imaging to measure interrecti distance. *The Journal of orthopaedic and sports physical therapy*. 2013;43(7):495.
7. Mota PGFD, Pascoal AGBA, Carita AIAD, Bø K. Prevalence and risk factors of diastasis recti

	abdominis from late pregnancy to 6 months postpartum, and relationship with lumbo-pelvic pain. Manual Therapy. 2015;20(1):200-5. 8. Lee D, Hodges PW. Behavior of the Linea Alba During a Curl-up Task in Diastasis Rectus Abdominis: An Observational Study. The Journal of orthopaedic and sports physical therapy. 2016;46(7):580.
--	---

VERSION 1 – AUTHOR RESPONSE

Reviewer 1

1) Title: Suggest adding “inter-rater” before reliability in title and aims.

Thank you for this comment. We agree that it is important to clearly state the type of reliability:

- We have changed the title to: Clinical assessment of pelvic floor and abdominal muscles three months postpartum: An inter-rater reliability study (page 1, line 1)

- We have changed the design to: A multi-center inter-rater reliability study (abstract, line 4)

- We have changed the objectives to: Evaluation of the inter-rater reliability of clinical assessment methods for pelvic floor muscles and diastasis recti abdominis postpartum (abstract, line 2-3)

- We have changed the objectives in the introduction to: “We aimed to evaluate the inter-rater reliability of different aspects of the clinical assessment of pelvic floor muscle and diastasis recti abdominis using observation, calipers, and palpation at three months postpartum. As you can see on page 5, line 18-20

2) Abstract, results, line 48: Add “at the umbilicus” as other ICC values were not quite as high.

Thank you for noticing that we missed this important information. We have added “at the umbilicus”, as you can see in the abstract, line 48.

3) Keywords: Recommend adding pelvic floor muscles

Thank you for this suggestion, we have added “pelvic floor muscles” to keywords, as you can see on page 3, line 16

4) The participating physical therapists had completed at least a 4-day course in pelvic floor muscle assessment and treatment, and all had 1-9 years’ experience in assessing these muscles. This is important information to place in the abstract, as it speaks to the generalizability of the findings. Consider adding to last sentence of ‘participants’: “At each center, 2 physiotherapists with training and experience in pelvic floor assessment assessed...” (To decrease words, you could delete “with the ability to understand and respond in Swedish”, as this is presented later.)

We agree that this is important information and have changed this paragraph in the abstract to:

“Eligibility for participation included female gender, ≥ 18 years, at maximum three months after childbirth. Exclusion criteria were chronic pelvic girdle pain and/or low back pain and/or pelvic floor tear grade III/IV. At each center, two physiotherapists, with training and experience in pelvic floor assessment, assessed the 222 women according to a standardized protocol in random order”, as you can see in the abstract, line 19-28,

5) Page 5, line 37: This should be “International Continence Society”

Thank you for noticing this mistake, we have changed to International Continence Society, as you can see on page 5, line 37.

6) Page 7, line 30: Please state the minimum number of days after delivery exams were undertaken (that is, “within 3 months” provides only the maximum).

We agree that this is important information and have changed to between eight to twelve weeks after giving birth, please see at page 6 line 20-21-

7) Page 7, line 36: “Plinth” is not a familiar word to me (and the definition in Webster’s is not likely what you mean!)—please find a substitute.

It is really important for us non-native speakers to learn the right vocabulary in this field. We found the word “plinth” in other articles about pelvic floor muscles and diastasis recti abdominis assessment. However, as you state, the definition Webster’s is not what we meant. We have changed to “The pelvic floor muscles were assessed with the participant in supine position on a flat bench.”, as you can see on page 8, line 7 and line 14+15.

8) Methods: Was the order of the PT exams at each site randomly assigned? If not, please state in results whether an order effect was seen, particularly for the modified Oxford scale.

We agree that this is important information and this has been clarified in the sentence: “The two physiotherapists at each site were assessing in random order, were blinded to each other’s findings, and not allowed to talk about their assessments”, please see page 8, line 24 and page 9, line 1.

9) Page 11 Table 1, line 30: 3% for center 3 for number of children=1 should be 30%

Thank you for noticing this mistake. We have changed to 11 participants (30%) as you can see in table 1.

10) It is true that reliability for diastasis recti width measurement was good (discussion, page 15, lines 30-34). However, Table 3 (page 13) raises the question of whether the measure is accurate. The mean DR width at center 3 was, as you note, significantly wider than other sites, with about a 1-cm difference

between center 3 and center 1 for all three locations. This may be due to the difference in population at center 3 (particularly notable is increased proportion of multiparas at center 3) or may be due to residual problems with test performance even after retraining.

It would be helpful to perform an exploratory analysis of mean width at each site in just the subgroup with 1 child and another in just the subgroup with 2 children to get a sense of whether the results seem more in line.

Thank you for raising this important question. To clarify this, we added the subgroups primipara and multipara to table 3 (as you can see on page 14). The data shows that women with more than one child have a significantly wider diastasis at the umbilicus at rehabilitation centre 1 but not at rehabilitation centre 2+3, which as you noted, can raise the question if the measurements are accurate. We discuss this at page 19, line 16-22.

11) Page 19, line 37-41: I did not see an analysis of differences according to age, BMI, etc in the results section.

Thank you for noticing this. We have changed to: "Thus, the descriptive statistics and the calculate statistically significant differences between the three rehabilitation centers regarding age, BMI, mode of delivery, pelvic floor tearing, and highest birth weight were based on self-reported data from the participants", as you can see on page 21, line 8-11, we also added the statistically significant differences to table 1 (page 10)

12) Page 20, line 43, conclusion: Probably "requesting" would be a better translation than "demanding".

Thank you for providing a better translation, we changed to: Women are increasingly requesting assessment of their pelvic floor muscles and diastasis recti abdominis after pregnancy, as you can see on page 22, line 21-22

13) Appendix: This information is very valuable and will be helpful to future researchers and clinicians. The photographs are helpful in understanding how measures were taken. For diastasis recti abdominis bulging, please describe the 3-step sit-up test, rather than simply providing a reference.

We agree that the clinical assessment procedure should be described as detailed as possible to make this article helpful for both researcher and clinicians. We have added "The participant was laying supine with straight legs on the flat bench. The test was rated as 0 if the participant was not able to perform a sit-up, as 1 if the participant was able to raise the upper torso to 40-degree angle from the bench with straight and secured legs, arms at the side of the body. The test was rated as 2 if the participant was able to perform the same task with the hands held behind the head and as 3 if the participant was able to perform a sit-up with the hands behind her head and hips and knees flexed and not secured. The participant had to hold the sit-up position for 5 seconds", as you can see in supplement 1.

Reviewer 2

Below we provide a point-by point response to your comments:

I recommend that it is clearly stated both in the abstract and the article what type of reliability was investigated.

Thank you for this comment. As the main focus is on inter-rater reliability, we agree that it should be more clearly stated, therefore:

- We have changed the title to: Clinical assessment of pelvic floor and abdominal muscles three months postpartum: An inter-rater reliability study (page 1, line 1)

- We have changed the design to: A multi-center inter-rater reliability study (abstract, line 4)

- We have changed the objectives to: Evaluation of the inter-rater reliability of clinical assessment methods for pelvic floor muscles and diastasis recti abdominis postpartum (abstract, line 2-3)

- We have changed the objectives in the introduction to: “We aimed to evaluate the inter-rater reliability of different aspects of the clinical assessment of pelvic floor muscle and diastasis recti abdominis using observation, calipers, and palpation at three months postpartum. As you can see on page 5, line 18-20

Methods regarding pelvic floor function are sufficiently described. The methods described to measure the diastasis rectus abdominis width is not sufficiently described. It is rather confusing to the reader whether the width has been measured during complete rest or during a head lift. In the existing literature the width of the diastasis rectus abdominis is either measured at rest, with the head resting on a plinth or a pillow, or at activation of the abdominal muscles, including a headlight. As far as I can interpret the assessment protocol of the diastasis rectus abdominis, the width was measured with a relaxed headlit. How can you then be sure that the abdominal musculature is actually at rest? And why do you choose a different measurement protocol to other studies. Please explain the rationale for this further, as it may imply that the results of this study cannot be compared to results from other similar studies (2-6).

Thank you for this very important comment. As you can see in supplement 1, we worked a lot with the right application and procedure of measuring the DRA width. As you stated, the existing studies about caliper measurement from Boxer and Jones¹ and Chiarello et al² were measuring in both resting and active position. However, in the training sessions to this study, we found it almost impossible to measure in resting, even if the above mentioned studies described the measurement in resting as reliable methods. At the same time, we could see that as higher the head lift/shoulder lift as less widths was measurable which is in line with a study showing that an abdominal crunch is narrowing the inter-recti distance.³ We have clarified our reasoning in supplement 1 on page 2: “With less lift, there was no contraction in the abdominal muscles and it was almost impossible to find the inner edges of the musculus rectus abdominis and with a higher lift the distance between the two bellies of the musculus recti abdominis was decreasing which is in line with the study of Mota et al showing that an abdominal crunch is narrowing the inter-recti distance.³”

Dalal et al⁴ and Parker et al⁵ described a measurement taken in a modified curl-up position (spine of the scapula off the table but not the inferior angle). They also used the concentric and eccentric movement of lowering the head to get better contact with the muscles, as we did in our study.

It is important to clarify that our intention was to measure in an active position but without missing the existing width between the muscle bellies. We have clarified the rationale in supplement 1 (“To avoid pre-activation of the deeper abdominal muscle, the participants were asked to relax before performing the trained head lift of 2–3 cm. During this movement -resulting in an activation of the musculus rectus abdominis- the physiotherapist identified the distance between the two parts of the rectus abdominis with her fingers and measured this felt distance using the caliper (Image 1a)”) and have also discussed the implication of this measurement protocol in the discussion, please see page 21, line 19-25.

On page 8, last paragraph, it is stated that the assessment of the diastasis recti abdominis was conducted with the participants in the same position as described for the assessment of the pelvic floor muscles. It is described that the head rests on a pillow. In the supplement however, it is stated that there is no pillow during the assessment. Please clarify. It is not described where the participants keep their arms during the assessment of the diastasis rectus abdominis. This is important as different arm positions may contribute to different recruitment of the abdominal muscles.

Thank you for drawing our attention to this confusing information. We have clarified this in the paragraph about diastasis recti abdominis assessment, please see page 8, line 14-16. "The assessment of the diastasis recti abdominis was conducted in supine position on a flat bench, without a pillow. The participants were in hook-lying position with their arms resting at their sides."

Regarding the assessment of the pelvic floor, it is described in the supplement that during observation of the pelvic floor muscle contraction, the physiotherapist stood beside the plinth, holding the participants legs...I assume this ought to be supporting the participants legs? This must be clarified.

Yes, you are right, and we have changed to "supporting the participant's legs with her hands and observing the movement of the perineum" (supplement 2) to clarify this.

The rater population ought to be more specified with regards to experience, training, work in this area etc, and explain the rationale for including only novice raters for the diastasis rectus abdominis measurements. Equally for the pelvic floor assessment, the different raters experiences may influence the results (1).

We agree that this is important information and we have provided a more detailed description and explanation of the physiotherapists' experience, as you can see on page 7, line 4-13: "The physiotherapists at primary care rehabilitation center 1 had two and nine years' of experience in pelvic floor muscle assessment at start of the study, the physiotherapists at primary care rehabilitation center 2 had one and three years' experience and the physiotherapists at primary care rehabilitation center 3 had both one year experience in pelvic floor muscle assessment at start of the study. All six physiotherapists work at primary care rehabilitation centers, part- of fulltime with women's health. They had experience in palpating the diastasis recti abdominis with the finger-width method. However, similar to the low numbers of American physiotherapists using the caliper,6 caliper measurement was new to all raters as it is unusual in Sweden and, diastasis recti abdominis assessment is not part of the Swedish physiotherapy education."

Was the final measure/rating a result of single or repeated measures? Were the raters blinded to the participants background information (Hawthorne effect)?

Thank you for this question, which we also discussed in the design phase of this study. We decided to base the final measures/ratings on single assessments. Our rationale for this was that there were many different aspects to be assessed, and we wanted to avoid a fatigue effect. We have clarified this on page 8, line 19. The raters were blinded to the participants background information and we have added this information on page9, line 2.

Are research ethics addressed appropriately?

Is this study in line with the Helsinki declaration? Also include what reporting guidelines were used when writing the manuscript.

Thank you for this comment. The study was approved by the ethical board in Sweden which is based on the ethical principles stated in the Helsinki declaration. We have added this to page, line 11-12: "This study is in line with the Helsinki declaration and the STARD (Standards for Reporting Diagnostic accuracy studies) checklist was used to report this study about assessment methods."

Are the references up to date and appropriate?

Page 5, 2nd paragraph line 7: Spelling mistake. It should be The International Continence Society (ICS). Not The international incontinence society.

Thank you for noticing this spelling mistake, we have changed " The International Continence Society", please see page 4, line 15.

Page 3-4, 1st paragraph: The function of the human core and how the different muscles interact and how this may affect function is under debate. I would recommend modifying and adding other references to the introduction. I also question the use of the reference of Mota et al (2015) when referring to postpartum muscular alterations can lead to feelings of insecurity. They found that women with diastasis rectus abdominis were not more likely to report lumbo pelvic pain than women without diastasis rectus abdominis (7).

Thank you for this input on right references, we agree that there is no current consent about core muscle interaction and their associations to function. Since core stability and muscular interactions are not the within the scope of this article, we choose to shorten the paragraph and put more emphasis on the paragraphs about measurement methods. The following change was made: "Both the pelvic floor and the abdominal muscles are part of the stabilization system for the pelvic and spine⁷ as well as the continence system.⁸ Pregnancy and childbirth cause alterations in these muscle groups. During a vaginal delivery, the pelvic floor muscles overstretches up to three times.⁹ It takes up to approximately six months until muscles, nerves and the connective tissue are recovered from a vaginal delivery.^{10,11}, please see page 3, line 23-24 and page 4 line 1-2.

-

The rationale to measure diastasis rectus abdominis depth is based on the study by Lee & Hodges (2016). They argue that the tension or depth of the Linea Alba may be used to assess linea alba behaviour in patients with diastasis rectus abdominis. However, they have based these theories on a measurement method which is not tested for validity nor reliability⁽⁸⁾. Will the depth of the diastasis rectus abdominis give information on the actual depth, or the behaviour and tension in the connective tissue linea alba?

This is a really good and interesting question. As discussed on page 20, the investigating physiotherapists hypothesized that "resistance in the depth" was felt and assessed as soon as the participants did not activate their deeper abdominal muscles during the head lift. If this is true, we were

actually investigating the ability to activate the musculus transversus abdominis during a head lift which increases the tension in the linea alba. In a next study, we could correlate the depth with the width with the hypothesis that the depth influences the width, knowing that a transversus abdominis activation is increasing the width¹²

We agree with you that we cannot base our studies on findings of studies with poor research quality. However, in clinical practice a lot of women are searching for help due to problems with their diastasis recti abdominis postpartum.⁶ As we see in research, there are no correlations or associations between diastasis recti abdominis width and lumbopelvic pain and incontinence.¹³ Our intention was to find easy applicable, inexpensive methods to assess other aspects of the diastasis recti abdominis. We have clarified this in the introduction on page 5, line 6-17.

As the authors of this manuscript states there are no validated and clinically applicable assessment methods or rating scales for the parameters of diastasis recti abdominis depth and bulging. I therefore question the validity of the depth measurement in this manuscript. How do the authors know that the raters are measuring the depth of the diastasis, and not the tension in the connective tissue? Physiotherapists in clinical practice do already assess the depth/tension, but we do not know how to interpret the results. I would welcome more information on what you assume you are measuring. Thank you for this important comment, we assume that the depth of the diastasis recti abdominis is the tension in the linea alba, which we have clarified in the introduction (page 5, line 15-17) and also in the supplement 1 (page 4): “palpated the tension in the linea alba (without adding pressure).”

When the results of the study show fair to moderate weighted values and the validity of the depth assessment is questionable, I would recommend avoiding conclusions that this is a method which may be used in clinical practice. If clinicians read more into this assessment method than what the evidence allow, it will certainly not improve the patient care.

We agree and have added “Before implications of this methods to clinical practice, further validation of the assessments of diastasis recti abdominis depth and bulging, for example a correlation between ultrasound/ shear-wave elastography and palpation, is needed” in the discussion (page 20, line 20-22)

I do not think this study adds two novel methods and scales for the assessment of the diastasis recti abdominis. More studies are needed to establish why we should measure depth/tension and bulging, and we need more reliable methods than these assessed here.

We agree and have planned further research in this area. We have changed this in the section strengths and limitations (page 3, line 11-12); “This study is raising novel thoughts about the question how diastasis recti abdominis depth and bulging could be tested postpartum.”, as well as in the abstract (page 3, line 2-3): “This study provides novel thoughts about how to measure diastasis recti abdominis depth and bulging.

VERSION 2 – REVIEW

REVIEWER	Nygaard, Ingrid University of Utah School of Medicine, OB/GYN
REVIEW RETURNED	24-Jun-2021

GENERAL COMMENTS	Thank you for the additional information you have provided in response to reviewers' comments.
--

REVIEWER	Theodorsen, N-M University of Bergen
REVIEW RETURNED	14-Jul-2021

GENERAL COMMENTS	The manuscript has improved greatly. However there are still some adjustments I would recommend First of all I want to clarify that I am not a Dr. And should not be titled as such. Comment one: 4. Are the methods described sufficiently in order to be repeated? I am still confused about the bulging measurements of the DRA described in supplement 1. You refer to the 3 step sit-up test. What results do you include in the table presenting the measurements of bulging? Ability to perform a sit-up? Presence of bulging in all participants, or only the ones that can perform a complete sit up? What is a bulging? And is an observed bulging in participants not able to perform a sit up the same as an observed bulging in a complete sit up? What I ask is Do you know what you are measuring? Comment two: Results are not presented clearly in the discussion section p 17, line 3-9, and also the Future research section p 22, line 8. Please clarify and be aware that you present results from both pelvic floor and DRA in a confusing manner. State clearly if you are referring to the pelvic floor or DRA. And I would recommend to continue as you have in the introduction to present pelvic floor first, and DRA secondly throughout the text. This will make it easier for the reader to follow. Comment three: Limitations. I am still concerned about the rationale, the method and the validity of measurements of the DRA depth in this study. What is the anatomical definition of the depth of the DRA? Are you measuring what you think you are measuring? If not, how will this manuscript influence clinicians? How do the authors know that the raters are measuring the depth of the DRA or the tension in the connective tissue? Why should clinicians measure depth and can this be done by palpation? Page 5, line 6: You state that one study SHOWS. I would recommend to change this with suggests, as the methods applied in this referred study has not been validated.
--

	Spelling:  1. The word plinth is the correct English term for flat bench used by physiotherapists. 2. P21, line 2. Do you mean implementation, not implications? 3. P 22, line 3 please rephrase this paragraph from "However" as it is somewhat confusing. 4. P 22, line 7: use at rest, not in resting 5. P22, line 8: we registered that the higher the head lift/shoulder lift the less width was measurable compared to at rest.....
--	---

VERSION 2 – AUTHOR RESPONSE

1. Are the methods described sufficiently in order to be repeated? I am still confused about the bulging measurements of the DRA described in supplement 1. You refer to the 3 step sit-up test:
 1. What results do you include in the table presenting the measurements of bulging? Ability to perform a sit-up?

Thank you for this question, for this paper, the sit-up test was used to assess the diastasis recti abdominis bulging during an exertion, and these assessments are presented as results in Table 5. The ability of performing a sit-up will be part of the analysis for a further study.

2. Presence of bulging in all participants, or only the ones that can perform a complete sit up?

We have clarified this in the Results section, as you can see on page 16, line 3-7. “The diastasis recti abdominis bulging was rated in all participants, even those who were not able to perform a complete sit-up. According to Lo et al, the diastasis recti abdominis bulging occurs on exertion.¹ Our hypothesis was that all participants did their individual maximal exertion to perform a sit-up even if it was not resulting into a complete sit-up. However, the assessing physiotherapist had the rating option “cannot assess” in case of insecurity.”

3. What is a bulging?

This was one of the main questions which we discussed when designing our study due the lack of a standardised definition. We have added the chosen definitions for diastasis recti abdominis width, depth and bulging in supplement 1

In surgical articles, both the protrusion/bulging of the whole abdominal wall ², as well as the midline abdominal bulge ³ are described. Lee *et al.* ⁴ differentiates between the bulging of the lower abdomen and the doming of the linea alba. The description of midline bulging during exertion, described by Lo et al.¹ is the one which is most often seen in clinical practice and the one which is used in our research.

4. And is an observed bulging in participants not able to perform a sit up the same as an observed bulging in a complete sit up?

This is a good question which we are not able to answer based on the result of the present study. As we see in studies about intraabdominal pressure in different movements, there are individual

variations in abdominal pressure in almost all exercises.⁵ As described above, we assumed an individual maximal exertion to perform a sit-up, and assessed the resulting midline bulge.

5. What I ask is Do you know what you are measuring?

We clarified in supplement 1, that we assessed the midline bulging during the attempt to do a sit-up (an exertion).

2. Results are not presented clearly in the discussion section p 17, line 3-9, and also the Future research section p 22, line 8. Please clarify and be aware that you present results from both pelvic floor and DRA in a confusing manner. State clearly if you are referring to the pelvic floor or DRA. And I would recommend to continue as you have in the introduction to present pelvic floor first, and DRA secondly throughout the text. This will make it easier for the reader to follow

Thank you for this suggestion to improve our article. We added the word “pelvic floor” in the Discussions section, page 16, line 22 and page 17, line 1 and re-wrote the section Future research. We have also included the arisen questions about DRA depth and bulging to the section “Future research”: Regarding the pelvic floor muscles, there remains a need for further research on how to assess and rate involuntary contraction, and voluntary relaxation, in the clinical assessment of women after pregnancy due to the high clinical relevance of this functions.

For the diastasis recti abdominis assessment, the depth and bulging have to be further defined and more knowledge about the behavior of the linea alba during muscle contraction and exertion is needed. For both pelvic floor muscle and diastasis recti abdominis assessment, we need to investigate the extent to which these values are clinically relevant for postpartum women—for example, if the assessment outcomes are associated with pain and dysfunction.

3. Limitations. I am still concerned about the rationale, the method and the validity of measurements of the DRA depth in this study.

1. What is the anatomical definition of the depth of the DRA?

Thank you for this comment, as mentioned above, we added in supplement 1 the chosen definitions for diastasis recti abdominis width, depth, and bulging.

2. Are you measuring what you think you are measuring?

Our hypothesis is that we are assessing the tension in the linea alba during an activation of the musculus rectus abdominis. In the design phase of this study, this test was added to the clinical assessment protocol of this study due to previous studies which discussed the influence of the deviant behaviour of the linea alba during the measurement of the diastasis recti abdominis width.^{6,7}

3. If not, how will this manuscript influence clinicians? How do the authors know that the raters are measuring the depth of the DRA or the tension in the connective tissue?

The palpation of the DRA depth is already a common part of the DRA assessment among physiotherapist working with women’s health. We agree that this assessment has to be critical discussed and our study about the inter-rater reliability was a first step. Further research about its

clinical relevance has to be done before clinical implementation, we added this important information to the conclusion of our paper: More research about the assessment of the involuntary contraction and voluntary relaxation of the pelvic floor muscles, as well as the diastasis recti abdominis depth and bulging is needed, before clinical implementation (page 23, line 8-11)

We cannot be sure that DRA depth is the tension in the linea alba. There are other hypotheses that DRA depth could be caused by a reduced solidity of the fibers in the linea alba and a greater laxity in the tissue.⁸ We discuss this on Page 20, line 15-19.

4. Why should clinicians measure depth and can this be done by palpation?

As we mentioned in our previous reply, in clinical practice a lot of women are searching for help due to problems with their diastasis recti abdominis postpartum. However, there are yet no shown correlations or associations between diastasis recti abdominis width and lumbopelvic pain and incontinence.⁹ Hypothetical, an altered function of the midline could lead to an abdominal trunk instability.¹⁰ The measurement of the DRA depth, defined as the palpated tension in the linea alba, could be an important new aspect due to the importance of the linea alba for the load transfer in the core. Based on our results, palpation of DRA depth is a questionable method due to its fair-to-moderate reliability, which we clarified p 20 line 19-21.

4. Page 5, line 6: You state that one study SHOWS. I would recommend to change this with suggests, as the methods applied in this referred study has not been validated.

Thank you for this comment, we have changed this on page 5 line 5: An experimental study suggests that the tendon between the two parts of the rectus abdominis—the linea alba— has less tension in a curl-up movement in women with diastasis recti abdominis.

Spelling:

1. The word plinth is the correct English term for flat bench used by physiotherapists.

As you can see in the original submitted document, we used the word plinth, however, reviewer 1 suggested this change. “Plinth” was according to her not a familiar word and had a different definition in Webster’s.

Thank you for correction of our spelling mistakes which were corrected as you can see in the red coloured text in the discussion section

VERSION 3 – REVIEW

REVIEWER	Theodorsen, N-M University of Bergen
REVIEW RETURNED	12-Aug-2021

GENERAL COMMENTS	It has been a pleasure to review your manuscript; it has been an educational process! Well done! I have one comment; one of the references has been listed twice in the reference list.
---